# Leveraging Image-to-Text Generators in Multimodal Vision Transformers for Inclusive Skin Cancer Diagnosis: A Comparative Study

Chentao Jin
*dept. Electrical and Computer Engineering*
*University of Waterloo*
Waterloo, Canada
c65jin@uwaterloo.ca

Eman Rezk
*dept. Electrical and Computer Engineering*
*University of Waterloo*
Waterloo, Canada
e2rezk@uwaterloo.ca

Walaa Moursi
*dept. Combinatorics and Optimization*
*University of Waterloo*
Waterloo, Canada
walaa.moursi@uwaterloo.ca

Zhou Wang
*dept. Electrical and Computer Engineering*
*University of Waterloo*
Waterloo, Canada
zhou.wang@uwaterloo.ca

*Abstract*—AI models for skin cancer diagnosis often underperform on darker skin tones due to imbalanced training datasets that predominantly feature lighter skin. In this study, we investigate whether lightweight, textual input can mitigate this disparity in a low-data setting. We use a dataset of only 4,311 clinical dermatology images—3,900 from lighter skin tones and just 411 from darker tones—to train Vision Transformers (ViTs) enhanced with text input including skin tone and generated lesion descriptions from Gemini and MONET. These textual inputs are fused with visual features via late fusion strategies. Among all configurations, ViT-B/32 combined with BERT-encoded skin tone using Element-Wise Fusion achieved the most balanced results, with AUCs of 0.822 (light) and 0.825 (dark), and matched accuracies of 0.823. This setup reduced the AUC gap to 0.003 and the accuracy gap to 0.0001. Our findings show that incorporating simple and domain-specific textual input can substantially reduce skin tone bias in ViT-based diagnosis offering a practical solution for building fairer medical AI.

*Index Terms*—skin cancer detection, vision transformers, multimodal learning, vision-language models.

## I. Introduction

SKIN cancer is a common and potentially fatal disease affecting all skin tones, though symptoms vary with pigmentation. Individuals with skin of color are more often diagnosed at later melanoma stages, contributing to lower survival rates [1]. These disparities challenge accurate diagnosis and are worsened by underrepresentation of darker skin tones in dermatology datasets. Machine learning models trained on such imbalanced data frequently show performance gaps, leading to misdiagnoses and deepening health inequities [2]. Ensuring fairness in AI-driven diagnosis is thus essential, particularly for tasks like skin cancer detection. While CNNs have achieved high lesion classification accuracy, they struggle with fairness due to data imbalance [3].

Vision Transformers (ViTs) improve on this by applying transformer architectures to images via patch tokenization and self-attention [4]. Their global attention captures both lesion structure and texture, supporting uniform focus across varied skin tones. As each patch acts as a token, ViTs allow for re-weighting underrepresented tones, potentially reducing bias.

This study evaluates ViTs ability to address skin tone disparities. We further explore multimodal input—combining images with skin tone and lesion descriptions—to boost fairness. Two image-to-text models are used: Gemini, a general-purpose model by Google [5], and MONET, a dermatology-specific model validated by experts [6]. Our contribution is as follows:

1) Investigate ViT, hybrid ViT with CNN, and pure CNN in reducing accuracy gap between light and dark skin tones.
2) Study the impact of using different text inputs on model performance across skin tones.
3) Utilize two image-to-text models for text generation.
4) Evaluate various text encoding and image and text fusion strategies.
5) Conduct skin tone-based analysis for all models.
6) Assess models using unbiased metrics based on per-group averaging.

*Code Base: https://github.com/GeekChentao/Debias-ViT-on-Dermatology-Images*

## II. Related Work

CNNs have been widely used for skin cancer diagnosis. For instance, a recent study showed that a CNN-based model achieved 80% accuracy in classifying malignant and benign skin lesions, demonstrating its effectiveness in melanoma diagnosis [7]. Similarly, an optimized CNN architecture achieved 97.86% accuracy across seven skin lesion types, including melanoma and basal cell carcinoma [8]. However, these models are susceptible to data-driven biases. Pope et al. [9] found that 83.3% of images in the ISIC archive represent light-toned

skin, leading to disparities in diagnostic accuracy across different skin types. Building on this, Benmalek et al. [3] confirmed that CNN (DenseNet) -based OOD models exhibit significant performance disparities when evaluated across different skin tone groups, with poorly performing models showing a 10-30% drop in performance for darker skin tones (Fitzpatrick types V–VI) compared to lighter ones (Fitzpatrick types I–IV).

Recent advancements have seen the integration of ViTs [4] into dermatological applications, particularly in skin cancer diagnosis. A systematic review by Adebiyi et al. [10] analyzed various transformer techniques applied to skin lesion classification and diagnosis, highlighting their superior performance in handling visual ambiguities and irregular lesion shapes compared to CNNs. Another study employing a ViT model on the HAM10000 dataset achieved a classification accuracy of 96.15%, outperforming several CNN-based approaches [11].

Integrating clinical metadata with dermoscopic images has shown promise in enhancing skin cancer diagnosis. Ou et al. [12] developed a deep learning model that combines clinical images and metadata, demonstrating improved diagnostic performance. Similarly, Tang et al. [13] introduced a fusion structure with a fusion attention module for multimodal skin cancer classification, achieving superior results by combining dermatological images and patient metadata.

In light of prior research demonstrating the strengths and limitations of CNNs, the emerging potential of ViTs, and the benefits of multimodal fusion, this paper aims to systematically evaluate ViT and multimodal architectures that combine a ViT for image encoding with a text-based transformer for text input integration (skin tone and lesion descriptions), to both improve accuracy and reduce the performance gap between skin tones in skin cancer diagnosis.

## III. DATASET

The Fitzpatrick 17k dataset [14], [15], is one of the largest publicly available dermatology image collections, containing 17,000 images across 114 skin conditions, annotated for condition, malignancy, and Fitzpatrick skin types (I–VI). Since our focus was malignancy detection, we selected only malignant and benign cases, resulting in 4,311 images (2,155 malignant and 2,156 benign). To analyze performance differences across light and dark skin tones, we followed Benmalek et al. [3] in grouping Fitzpatrick Types I–IV as Skin Tone 1 (light tones) and Types V–VI as Skin Tone 2 (dark tones), yielding 3,900 and 411 images, respectively. Specifically, Skin Tone 1 includes 1,947 malignant and 1,953 benign cases, while Skin Tone 2 comprises 208 malignant and 203 benign cases.

Alongside image data, we included textual inputs to evaluate their impact on improving overall accuracy and fairness in predictions across skin tones. Specifically, we used skin tone (from Fitzpatrick annotations) and lesion descriptions generated by the Gemini and MONET models.

Gemini is a general-purpose multimodal generative model [5], which we used solely for image-to-text generation to produce lesion descriptions. To adapt it for this task, we applied few-shot learning with 32 image-description pairs

from SKINCON [16], a dataset containing dermatology images labeled with concepts such as ulcer and crust. These examples guided Gemini in generating consistent, domain-relevant descriptions for images in the Fitzpatrick 17k dataset. An example output is shown in Figure 1. We then created **Gemini Lesion Description** by appending the Gemini output to the corresponding skin tone, producing a description in the format: `{skin tone} {Gemini description}`.

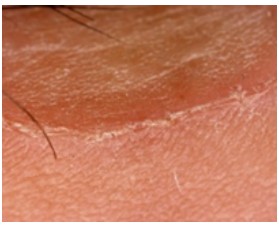

Fig. 1: An image from the dataset, Gemini-generated description: "The image shows a close-up of skin with dry, flaky texture. There are visible cracks and lines in the skin surface."

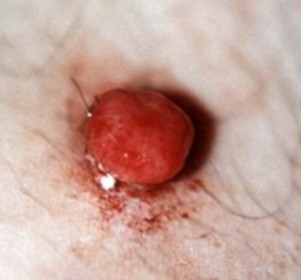

Fig. 2: Abscess presence score generated by MONET.

MONET is a domain-specific model trained on $105\,550$ dermatological images and their textual descriptions, developed to accurately annotate lesion concepts in dermatology images [6]. Its effectiveness has been validated by board-certified dermatologists and shown to be competitive with supervised models trained on concept-annotated clinical datasets. For each lesion concept (e.g., "Abscess"), MONET generates a confidence score between 0.0 and 1.0 indicating its presence in an image. Figure 2 shows examples of image–score pairs. We used MONET to generate scores for 48 lesion concepts per image, then constructed a structured textual description—**MONET Full Lesion Description**—in the format: `{skin_tone} {concept 1}:{score} present, {concept 2}:{score} present`, etc.

While the full MONET Full Lesion Description covers 48 lesion concepts and offers a comprehensive overview, it often includes low-relevance details, such as concepts with presence scores below 0.2. Incorporating the full set may introduce noise and reduce the efficiency of downstream tasks. To address this, we constructed a simplified version, referred to as the **MONET Top-3 Concepts Lesion Description**, by pruning the concepts with less weights and retaining only the three concepts with the highest presence scores per image. The resulting format is: `{skin tone} {concept with the highest weight}, {concept with the second-highest weight}`.

This compact representation enables the model to focus more effectively on the most salient lesion features.

## IV. METHODOLOGY

### A. Vision Model Architectures

In this work, we utilized **ViT-B/32** due to its computational efficiency. Larger variants, such as **ViT-L/32** and **ViT-H/14**, were not considered for multimodal evaluation due to their significantly higher memory and training demands. ViT-B/32 processes images in the following steps:

- **Patch Extraction:** The input image is divided into fixed-size non-overlapping patches (32×32 pixels).
- **Linear Projection (Patch Embedding):** Each patch is flattened and linearly projected using a fully connected layer into a D-dimensional embedding ($D = 768$).
- **Positional Encoding:** A learnable positional embedding is added to retain spatial information.
- **Transformer Encoder:** The sequence of patch embeddings is then fed into a standard transformer encoder (multi-head self-attention and MLP layers).
- **Classification Token:** A special token is prepended, and its final embedding is used for classification.

Alongside ViT-B/32, we implemented a hybrid model-**ResNet-ViT-B/32**-which combines ViT with ResNet-26 to improve efficiency. Rather than applying patch tokenization to raw pixels, this model uses convolutional layers to extract features, then applies tokenization on the resulting feature maps. Wahid et al. [17] showed that combining ResNet and ViT enhanced feature extraction, improving accuracy in myocardial infarction detection. This architecture has the potential to generalize across subgroups by emphasizing structure while mitigating bias. Both ViT-B/32 and ResNet-ViT-B/32 were trained on image-only data to serve as visual baselines.

### B. Text Encoding and Integration Approaches

Building on a strong vision model, we enhanced its capability by incorporating supplementary text—specifically, skin tone and lesion descriptions from Gemini and MONET—to reduce bias and improve performance across skin tones. Textual inputs were processed using one of three pre-trained text transformers—**CLIP** [18], **BERT** [19], or **Sentence-BERT** (**S-BERT**) [20]—each paired with its respective tokenizer. CLIP aligns image and text embeddings in a shared latent space, making it effective for multimodal tasks. BERT generates contextualized word embeddings but requires additional pooling (e.g., using the [CLS] token) to obtain fixed-size outputs. S-BERT is a fine-tuned BERT variant that produces dense sentence embeddings optimized for semantic similarity.

Our multimodal architecture employed a late fusion strategy, which had been shown to outperform early fusion in various image-text tasks [21]. We implemented two late fusion approaches. In **Concatenation Fusion**, the text embedding was appended to the ViT's [CLS] token output, and the resulting vector was fed into a classification sub-network. In **Element-Wise Fusion**, both image and text embeddings were projected to the same dimension and merged through element-wise operations before being passed to the final classification layer. By comparing these strategies, we aimed to identify the configuration that most effectively balanced predictive performance and fairness across skin tones.

Figure 3 presents an overview of our experimental framework, including vision-only baselines and multimodal architectures. The vision baselines consist of **ViT-B/32**, **ResNet-ViT-B/32**, and a 16-layer **CNN**, all trained solely on lesion images. For the multimodal setups, we incorporated 4 types of textual inputs: (1) **skin tone**, (2) **Gemini Lesion Description**, (3) **MONET Full Lesion Description**, and (4) **MONET Top-3 Concept Description**. These were encoded using **CLIP**, **BERT**, or **S-BERT**, and fused with visual features via either **Concatenation** or **Element-Wise Fusion** strategies. We implemented a variety of model combinations illustrated in the figure to comprehensively evaluate performance across these configurations. Detailed experimental setups and results are presented in Section V.

## V. EXPERIMENTS

### A. Experimental Setup and Implementation Details

We optimized all models using stochastic gradient descent with a learning rate of 0.001 and a cosine annealing scheduler. The hyperparameters were chosen based on experiments with single-modality vision models, where they achieved the best performance. Data was split into 70% training, 15% validation, and 15% testing. Performance was evaluated using accuracy, F1 score, sensitivity (malignant detection), specificity (benign detection), and AUC. Metrics were computed separately for each skin tone, then averaged to assess overall performance; differences between the groups were also calculated to quantify performance gaps. All models were implemented in PyTorch and trained on Ubuntu 20.04.6 LTS with an NVIDIA GeForce RTX 3070 GPU.

### B. Vision Only Experimental Setup

The results of these experiments are summarized in Table Table I. Among the vision-only models, ViT-B/32 outperformed both the ResNet-augmented ViT-B/32 and the baseline 16-layer CNN, delivering more balanced classification across skin tone groups. ViT-B/32 achieved an average AUC of **0.807** with a performance gap of only **0.001**, and for the dark skin group, an AUC of **0.809** with a gap of **0.005**.

In comparison, ViT-B/32 with ResNet-26 exhibited larger disparities despite the added convolutional layers—yielding an average AUC of **0.763** with a gap of **0.040**, and an average accuracy of **0.736** with a gap of **0.041**. The baseline CNN, as expected given its simpler structure, performed the worst, with an average accuracy of **0.673** (gap: **0.056**) and an average AUC of **0.676** (gap: **0.053**).

These results indicated that the addition of ResNet preprocessing in the ViT-B/32 with ResNet-26 did not lead to performance improvement. Therefore, we selected **ViT-B/32** as the foundation for subsequent multimodal experiments.

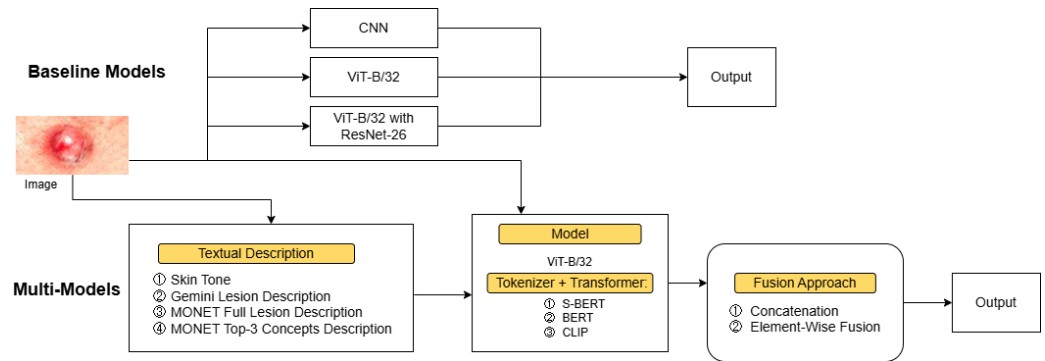

Fig. 3: Experimental workflow showing baseline model selection and multimodal integration using various text types, transformers, and integration strategies for skin cancer classification.

TABLE I: *Vision-Only Models Performance Evaluation*

| Model | Group | Accuracy | F1 Score | Sensitivity | Specificity | AUC |
|---|---|---|---|---|---|---|
| CNN | Light / Dark | 0.701 / 0.645 | 0.716 / 0.669 | 0.767 / 0.576 | 0.638 / 0.724 | 0.703 / 0.650 |
| | Average / Gap | **0.673 / 0.056** | 0.693 / 0.048 | 0.672 / 0.192 | 0.681 / 0.087 | **0.676 / 0.053** |
| ViT-B/32 | Light / Dark | 0.807 / 0.807 | 0.800 / 0.800 | 0.785 / 0.727 | 0.829 / 0.897 | 0.807 / 0.812 |
| | Average / Gap | **0.807 / 0.001** | 0.800 / 0.000 | 0.756 / 0.057 | 0.863 / 0.068 | **0.809 / 0.005** |
| ViT-B/32 with ResNet-26 | Light / Dark | 0.783 / 0.742 | 0.776 / 0.750 | 0.764 / 0.727 | 0.802 / 0.759 | 0.783 / 0.743 |
| | Average / Gap | **0.763 / 0.041** | 0.763 / 0.026 | 0.746 / 0.037 | 0.780 / 0.043 | **0.763 / 0.040** |

TABLE II: *Multimodal using skin tone text experimental setup*

| Model | Group | Accuracy | F1 Score | Sensitivity | Specificity | AUC |
|---|---|---|---|---|---|---|
| ViT-B/32 & S-BERT - Concat. Fusion | Light / Dark | 0.783 / 0.807 | 0.778 / 0.800 | 0.771 / 0.727 | 0.795 / 0.897 | 0.783 / 0.812 |
| | Average / Gap | 0.795 / 0.023 | 0.789 / 0.022 | 0.749 / 0.044 | 0.846 / 0.101 | 0.798 / 0.029 |
| ViT-B/32 & BERT - Concat. Fusion | Light / Dark | 0.795 / 0.807 | 0.790 / 0.807 | 0.781 / 0.758 | 0.809 / 0.862 | 0.795 / 0.810 |
| | Average / Gap | 0.801 / 0.011 | 0.798 / 0.017 | 0.769 / 0.024 | 0.835 / 0.053 | 0.802 / 0.015 |
| ViT-B/32 & CLIP - Concat. Fusion | Light / Dark | 0.833 / 0.807 | 0.827 / 0.807 | 0.813 / 0.758 | 0.852 / 0.862 | 0.832 / 0.810 |
| | Average / Gap | 0.820 / 0.026 | 0.817 / 0.020 | 0.785 / 0.055 | 0.857 / 0.010 | 0.821 / 0.023 |
| ViT-B/32 & S-BERT - Elem. Wise Fusion | Light / Dark | 0.809 / 0.807 | 0.804 / 0.807 | 0.795 / 0.758 | 0.822 / 0.862 | 0.809 / 0.810 |
| | Average / Gap | 0.808 / 0.002 | 0.805 / 0.003 | 0.776 / 0.038 | 0.842 / 0.040 | 0.809 / 0.001 |
| ViT-B/32 & BERT - Elem. Wise Fusion | Light / Dark | 0.823 / 0.823 | 0.819 / 0.825 | 0.816 / **0.788** | 0.829 / **0.862** | 0.822 / 0.825 |
| | Average / Gap | **0.823 / 0.0001** | 0.822 / 0.007 | 0.802 / 0.028 | 0.846 / 0.033 | **0.824 / 0.003** |
| ViT-B/32 & CLIP - Elem. Wise Fusion | Light / Dark | 0.841 / 0.774 | 0.838 / 0.774 | 0.837 / 0.727 | 0.846 / 0.828 | 0.841 / 0.777 |
| | Average / Gap | 0.808 / 0.067 | 0.806 / 0.064 | 0.782 / 0.110 | 0.837 / 0.018 | 0.809 / 0.064 |

## C. Multimodal using Skin Tone Text Experimental Setup

Table II summarizes the results. Among all configurations, **ViT-B/32 & BERT** with **Element-Wise Fusion** achieved the best overall performance and fairness across skin tones. Compared to the ViT-B/32 baseline—which reached an AUC of 0.807 for light skin and 0.812 for dark skin (gap: 0.005) and an average accuracy of 0.807 for both groups (gap: 0.001)—the multimodal setup improved AUCs to **0.822** for light skin and **0.825** for dark skin, reducing the AUC gap to just **0.003**. Accuracy increased to **0.823** for both groups with the gap remaining at **0.0001**. These gains did not compromise fairness, the model maintained high specificity (**0.862**) for dark skin while improving sensitivity (**0.788**), indicating better balance between true positives and true negatives.

These findings suggest that even minimal textual input—such as skin tone labels—when integrated via a robust transformer like BERT with Element-Wise Fusion, can enhance both fairness and predictive performance.

## D. Multimodal using Gemini Lesion Description Experimental Setup

As shown in Table III, incorporating Gemini Lesion Description resulted in a performance drop relative to the image-only ViT-B/32 baseline. The best-performing setup—**ViT-B/32 & CLIP Text Transformer** with **Element-Wise Fusion**—achieved AUCs of **0.827** for light skin and **0.795** for dark skin, yielding an average AUC of **0.811** and a gap of **0.033**. While the light skin AUC slightly surpassed the baseline (0.812), the decrease in dark skin AUC indicated reduced generalization and fairness. This configuration also underperformed compared to ViT-B/32 & BERT with Element-Wise Fusion incorporating skin tone which achieved higher and more balanced metrics (accuracy: 0.823 for both groups; AUC: 0.822 light, 0.825 dark).

Upon closer inspection of the generated descriptions, the observed performance decline stems from Gemini's few-shot lesion outputs lacking domain-specific adaptation. This limitation resulted in false, misleading, or irrelevant content (as shown in Figure 4). Further analysis of the token-level

TABLE III: *Multimodal using Gemini Lesion Description experimental setup*

| Model | Group | Accuracy | F1 Score | Sensitivity | Specificity | AUC |
|---|---|---|---|---|---|---|
| ViT-B/32 & S-BERT - Concat. Fusion | Light / Dark | 0.795 / 0.807 | 0.775 / 0.807 | 0.719 / 0.758 | 0.869 / 0.862 | 0.794 / 0.810 |
| | Average / Gap | 0.801 / 0.011 | 0.791 / 0.031 | 0.738 / 0.039 | 0.866 / 0.007 | 0.802 / 0.016 |
| ViT-B/32 & BERT - Concat. Fusion | Light / Dark | 0.809 / 0.774 | 0.808 / 0.781 | 0.816 / 0.758 | 0.802 / 0.793 | 0.809 / 0.775 |
| | Average / Gap | 0.792 / 0.035 | 0.794 / 0.026 | 0.787 / 0.058 | 0.798 / 0.009 | 0.792 / 0.034 |
| ViT-B/32 & CLIP - Concat. Fusion | Light / Dark | 0.826 / 0.790 | 0.821 / 0.787 | 0.813 / 0.727 | 0.839 / 0.862 | 0.826 / 0.795 |
| | Average / Gap | 0.808 / 0.036 | 0.804 / 0.034 | 0.770 / 0.085 | 0.851 / 0.023 | 0.810 / 0.031 |
| ViT-B/32 & S-BERT - Elem. Wise Fusion | Light / Dark | 0.804 / 0.758 | 0.792 / 0.762 | 0.760 / 0.727 | 0.846 / 0.793 | 0.803 / 0.760 |
| | Average / Gap | 0.781 / 0.046 | 0.777 / 0.030 | 0.744 / 0.033 | 0.819 / 0.053 | 0.782 / 0.043 |
| ViT-B/32 & BERT - Elem. Wise Fusion | Light / Dark | 0.804 / 0.774 | 0.791 / 0.767 | 0.757 / 0.697 | 0.849 / 0.862 | 0.803 / 0.780 |
| | Average / Gap | 0.789 / 0.030 | 0.779 / 0.025 | 0.727 / 0.060 | 0.856 / 0.013 | 0.791 / 0.024 |
| ViT-B/32 & CLIP - Elem. Wise Fusion | Light / Dark | 0.828 / 0.790 | 0.823 / 0.787 | 0.813 / 0.727 | 0.842 / 0.862 | 0.827 / 0.795 |
| | Average / Gap | **0.809 / 0.037** | 0.805 / 0.036 | 0.770 / 0.085 | 0.852 / 0.020 | **0.811 / 0.033** |

TABLE IV: *Multimodal using MONET Full Lesion Description experimental setup*

| Model | Group | Accuracy | F1 Score | Sensitivity | Specificity | AUC |
|---|---|---|---|---|---|---|
| ViT-B/32 & S-BERT - Concat. Fusion | Light / Dark | 0.807 / 0.774 | 0.793 / 0.774 | 0.743 / 0.727 | 0.869 / 0.828 | 0.806 / 0.777 |
| | Average / Gap | 0.791 / 0.033 | 0.784 / 0.019 | 0.735 / 0.016 | 0.848 / 0.042 | 0.792 / 0.029 |
| ViT-B/32 & BERT - Concat. Fusion | Light / Dark | 0.794 / 0.726 | 0.775 / 0.702 | 0.722 / 0.606 | 0.862 / 0.862 | 0.792 / 0.734 |
| | Average / Gap | 0.760 / 0.068 | 0.738 / 0.073 | 0.664 / 0.116 | 0.862 / 0.000 | 0.763 / 0.058 |
| ViT-B/32 & CLIP - Concat. Fusion | Light / Dark | 0.836 / 0.790 | 0.829 / 0.787 | 0.809 / 0.727 | 0.862 / 0.862 | 0.836 / 0.795 |
| | Average / Gap | **0.813 / 0.046** | 0.808 / 0.042 | 0.768 / 0.082 | 0.862 / 0.000 | **0.815 / 0.041** |
| ViT-B/32 & S-BERT - Elem. Wise Fusion | Light / Dark | 0.777 / 0.839 | 0.757 / 0.839 | 0.708 / 0.788 | 0.842 / 0.897 | 0.775 / 0.842 |
| | Average / Gap | 0.808 / 0.062 | 0.798 / 0.082 | 0.748 / 0.080 | 0.869 / 0.054 | 0.809 / 0.067 |
| ViT-B/32 & BERT - Elem. Wise Fusion | Light / Dark | 0.799 / 0.790 | 0.808 / 0.800 | 0.861 / 0.788 | 0.738 / 0.793 | 0.800 / 0.791 |
| | Average / Gap | 0.794 / 0.008 | 0.804 / 0.008 | 0.825 / 0.073 | 0.766 / 0.055 | 0.795 / 0.009 |
| ViT-B/32 & CLIP - Elem. Wise Fusion | Light / Dark | 0.843 / 0.758 | 0.833 / 0.746 | 0.795 / 0.667 | 0.889 / 0.862 | 0.842 / 0.764 |
| | Average / Gap | 0.801 / 0.085 | 0.789 / 0.087 | 0.731 / 0.128 | 0.876 / 0.027 | 0.803 / 0.078 |

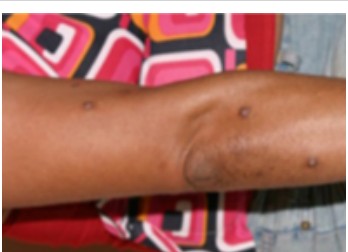

Fig. 4: Gemini generated description: "Multiple red papules on the arms", yet from this image, no red papules are visible.

attention weights from the text transformer revealed that a significant portion of attention was allocated to non-diagnostic elements—such as punctuation ([CLS], .) and common structural words ("the", "a", etc.). These findings highlight that while textual integration has the potential to improve fairness, the effectiveness of such inputs depends heavily on their clinical relevance and accuracy.

*E. Multimodal using MONET Full Lesion Description Experimental Setup*

As illustrated in Table IV, the **ViT-B/32 & CLIP** with **Concatenation Fusion** configuration yielded the highest average accuracy (**0.813**) and AUC (**0.815**) among MONET Full Description setups, though with notable performance gaps—**0.046** in accuracy and **0.041** in AUC. While these averaged metrics exceeded those of the baseline ViT-B/32, the increased disparity indicated reduced fairness. Moreover, this setup still underperformed relative to the ViT-B/32 with BERT and Element-Wise Fusion using skin tone, which achieved both higher and more balanced results (accuracy: 0.823 for both groups; AUC: 0.822 light, 0.825 dark).

While the MONET Full Lesion Description is more structured, accurate, and detailed than Gemini's, its inclusion of 48 lesion concepts per image adds excessive details. This abundance of information can act as noise, potentially overwhelming the model and hindering performance in a low-data setting. Moreover, despite MONET's descriptions being validated by experts, they may still include misleading or inaccurate information. As reported in its original paper, MONET's performance varies across image types, achieving an AUROC of 0.767 on clinical images [6]. Therefore, a more concise and targeted format—MONET Top-3 Concepts Lesion Description—is preferable for improving information density and accuracy and supporting better model performance.

*F. Multimodal using MONET Top-3 Concepts Lesion Description Experimental Setup*

As shown in Table V, the best-performing model in this group was **ViT-B/32** with **CLIP** and **Element-Wise Fusion**. This setup improved the average AUC from 0.815 (gap: 0.041) in the best MONET Full Description model to **0.826** (gap: **0.028**), and increased average accuracy from 0.813 (gap: 0.046) to **0.824** (gap: **0.035**). These gains were driven by the higher information density and accuracy achieved by retaining only the top three lesion concepts per image.

The most balanced performance was achieved by **ViT-B/32** with **CLIP** and **Concatenation Fusion**, which yielded identical AUCs of **0.814** for both light and dark skin tones—resulting in a minimal gap of **0.0001**. Nonetheless, both

TABLE V: Multimodal using MONET Top-3 Concepts Lesion Description experimental setup

| Model | Group | Accuracy | F1 Score | Sensitivity | Specificity | AUC |
|---|---|---|---|---|---|---|
| ViT-B/32 & S-BERT - Concat. Fusion | Light / Dark | 0.806 / 0.758 | 0.792 / 0.769 | 0.754 / 0.758 | 0.856 / 0.759 | 0.805 / 0.758 |
| | Average / Gap | 0.782 / 0.047 | 0.781 / 0.023 | 0.756 / 0.004 | 0.807 / 0.097 | 0.781 / 0.047 |
| ViT-B/32 & BERT - Concat. Fusion | Light / Dark | 0.809 / 0.807 | 0.804 / 0.813 | 0.799 / 0.788 | 0.819 / 0.828 | 0.809 / 0.808 |
| | Average / Gap | 0.808 / 0.002 | 0.808 / 0.008 | 0.793 / 0.011 | 0.823 / 0.009 | 0.808 / 0.001 |
| ViT-B/32 & CLIP - Concat. Fusion | Light / Dark | 0.816 / 0.807 | 0.791 / 0.793 | 0.708 / 0.697 | 0.920 / 0.931 | 0.814 / 0.814 |
| | Average / Gap | 0.811 / 0.009 | 0.792 / 0.002 | 0.703 / 0.011 | 0.925 / 0.012 | **0.814 / 0.0001** |
| ViT-B/32 & S-BERT - Elem. Wise Fusion | Light / Dark | 0.792 / 0.790 | 0.781 / 0.794 | 0.754 / 0.758 | 0.829 / 0.828 | 0.791 / 0.793 |
| | Average / Gap | 0.791 / 0.001 | 0.787 / 0.013 | 0.756 / 0.004 | 0.828 / 0.001 | 0.792 / 0.001 |
| ViT-B/32 & BERT - Elem. Wise Fusion | Light / Dark | 0.807 / 0.774 | 0.793 / 0.774 | 0.750 / 0.727 | 0.862 / 0.828 | 0.806 / 0.777 |
| | Average / Gap | 0.791 / 0.033 | 0.783 / 0.019 | 0.739 / 0.023 | 0.845 / 0.035 | 0.792 / 0.029 |
| ViT-B/32 & CLIP - Elem. Wise Fusion | Light / Dark | 0.841 / 0.807 | 0.826 / 0.800 | 0.764 / 0.727 | 0.916 / 0.897 | 0.840 / 0.812 |
| | Average / Gap | **0.824 / 0.035** | 0.813 / 0.026 | 0.746 / 0.037 | 0.906 / 0.020 | **0.826 / 0.028** |

TABLE VI: *Fairness Evaluation under Distribution Shift*

| Model | Group | Accuracy | F1 Score | Sensitivity | Specificity | AUC |
|---|---|---|---|---|---|---|
| ViT-B/32 & BERT - Elem. Wise Fusion | Light / Dark | **0.823 / 0.823** | 0.800 / 0.800 | 0.816 / 0.788 | 0.829 / 0.862 | **0.822 / 0.825** |
| | Average / Gap | **0.823 / 0.0001** | 0.800 / 0.000 | 0.802 / 0.028 | 0.846 / 0.033 | **0.824 / 0.003** |
| ViT-B/32 & BERT - Elem. Wise Fusion (under dist. shift) | Light / Dark | **0.811 / 0.823** | 0.805 / 0.813 | 0.813 / 0.818 | 0.809 / 0.828 | **0.811 / 0.823** |
| | Average / Gap | **0.817 / 0.012** | 0.809 / 0.007 | 0.815 / 0.006 | 0.818 / 0.019 | **0.817 / 0.012** |
| Single CNN | Light / Dark | 0.701 / 0.645 | 0.716 / 0.633 | 0.767 / 0.576 | 0.638 / 0.724 | 0.703 / 0.650 |
| | Average / Gap | **0.673** / 0.056 | 0.675 / 0.083 | 0.672 / 0.192 | 0.681 / 0.087 | **0.676** / 0.053 |
| Single CNN (under dist. shift) | Light / Dark | 0.664 / 0.613 | 0.669 / 0.586 | 0.691 / 0.515 | 0.638 / 0.724 | 0.664 / 0.620 |
| | Average / Gap | **0.638** / 0.051 | 0.628 / 0.083 | 0.603 / 0.176 | 0.681 / 0.087 | **0.642** / 0.044 |

models outperformed the overall best configuration: ViT-B/32 with BERT and Element-Wise Fusion using skin tone input, which achieved higher and more consistent results (accuracy: 0.823 for both groups; AUC: 0.822 light, 0.825 dark).

Even after improving information density by pruning infrequent concepts and retaining only skin tone and the top three lesion concepts, the model failed to utilize the additional input as effectively as it did with skin tone alone. The decline in performance is primarily due to input sparsity: selecting 3 out of 48 lesion concepts yields 17,296 unique combinations, which increases to 34,592 when including binary skin tone. Given the dataset contains only 4,311 images, most descriptions are either unique or occur infrequently, preventing the model from learning stable associations. These sparse combinations reduce information density and function as noise, thereby limiting the model's generalization capability.

### G. Best model Performance under Distribution Shift

We identified **ViT-B/32** & **BERT** with **Element-Wise Fusion** using skin tone as the best-performing model. To evaluate its generalizability and robustness, we tested it under a distribution shift using test-time augmented data simulating external datasets. This is particularly important for fairness-sensitive applications, where such perturbations may disproportionately affect underrepresented groups, such as individuals with dark skin. The augmentation pipeline included random horizontal flipping, color jittering (brightness, contrast, saturation), random erasing, and additive Gaussian noise to mimic real-world variations in lighting, occlusion, and visual noise.

As illustrated in the Table VI under perturbed conditions, the model's accuracy on light skin decreased slightly from 0.823 to **0.811**, while accuracy on dark skin remained stable at **0.823**.

This introduced a small accuracy gap of **0.012**, compared to the previously negligible gap of 0.0001. Similarly, AUC scores dropped modestly from 0.822 (light) and 0.825 (dark) to **0.811** and **0.823**, increasing the AUC gap from 0.003 to **0.012**. The average accuracy declined by **0.006** (from 0.823 to 0.817), and average AUC dropped by **0.007** (from 0.824 to 0.817), reflecting only minor performance degradation and strong robustness. In contrast, the simple CNN baseline experienced a more substantial drop in accuracy of **0.036** (from 0.673 to 0.637) and an AUC decline of **0.034** (from 0.676 to 0.642).

These comparisons underscore the robustness of the best-performing multimodal model, which demonstrated strong resilience to input perturbations while preserving both predictive performance and fairness across skin tones.

### H. Deployment Feasibility

Ensuring deployment feasibility is essential for translating medical AI models into practice. We evaluated several metrics—parameter count, training time, inference latency, and GPU/CPU memory usage—across six multimodal models incorporating skin tone text, and compared them against CNN and ViT-B/32 baselines.

As shown in Table VII, the best-performing model—**ViT-B/32** & **BERT** with **Element-Wise Fusion**—requires more resources compared to the baselines. Parameters increased from **25.8M** (CNN) and **87.5M** (ViT-B/32) to **198.3M**, and training time rose from **104.75s** (CNN) and **194.50s** (ViT-B/32) to **247.86s**. Inference latency remained efficient, increasing slightly from **1.69** ms/sample (CNN) and **1.63** ms/sample (ViT-B/32) to **2.02** ms/sample. GPU memory usage rose by about **1.3** GB, while CPU usage remained comparable.

TABLE VII: Computing Resource Comparison

| Model | Total Parameters | Inference Latency (ms/sample) | Training Time | MAX GPU Allocation (MB) | MAX CPU Allocation (MB) |
|---|---|---|---|---|---|
| **CNN** | **25,784,578** | **1.69** | **104.75** | **329.72** | 1244.03 |
| **ViT-B/32** | **87,456,770** | **1.63** | **194.50** | **1041.58** | 1154.50 |
| ViT-B/32 & S-BERT - Concat. Fusion | 110,759,810 | 1.84 | 191.87 | 1307.13 | 1421.38 |
| ViT-B/32 & BERT - Concat. Fusion | 197,725,442 | 1.94 | 274.86 | 2303.34 | 1513.28 |
| ViT-B/32 & CLIP - Concat. Fusion | 525,826 | 1.79 | 458.52 | 1381.44 | 1739.86 |
| ViT-B/32 & S-BERT - Elem. Wise Fusion | 110,759,810 | 2.06 | 159.40 | 1307.13 | 1526.75 |
| **ViT-B/32 & BERT - Elem. Wise Fusion** | **198,253,314** | **2.02** | **247.86** | **2314.21** | 1502.38 |
| ViT-B/32 & CLIP - Elem. Wise Fusion | 1,053,698 | 1.95 | 262.07 | 1389.64 | 1721.22 |

Despite the rise in computational demands, the increases are well within the limits of modern edge devices, suggesting that the proposed model is practically deployable.

## VI. Conclusion

We investigated fairness in skin cancer diagnosis by integrating vision and text-based transformers in a multimodal framework. Starting from a ViT-B/32, we evaluated multimodal setups combining image features with four text types: skin tone, Gemini Lesion Description, MONET Full Lesion Description, and MONET Top-3 Concepts Lesion Description. We also evaluated multiple text transformers and fusion strategies. Among all configurations, the model **ViT-B/32 & BERT** with **Element-Wise Fusion** incorporating skin tone text delivered the best results, achieving high accuracy while minimizing skin tone disparity. This suggests that combining vision transformers with well-structured textual input improves fairness in medical AI. However, the effectiveness of this architecture depends heavily on the accuracy, information density, and sparsity of the textual input.

## Acknowledgment

This work is supported in part by the Natural Sciences and Engineering Research Council of Canada (NSERC) Postdoctoral Award, Discovery Grant, Canada Research Chair Programs, and Ontario Early Researcher Award.

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
