# OpenReview forum: "Leveraging Image-to-Text Generators in Multimodal Vision Transformers for Inclusive Skin Cancer Diagnosis: A Comparative Study"
_IEEE.org/EMBS/BHI/2025/Conference — BHI 2025_

### Official Review · Reviewer_4Yw1 · 2025-07-13
**Leveraging Image-to-Text Generators in Multimodal Vision Transformers for Inclusive Skin Cancer Diagnosis: A Comparative Study**

**Confidence:** 4
**Clarity Of Writing:** great
**Clinical Significance:** excellent
**Methodological Novelty:** good
**Overall Rating:** 6
**Final Rating:** 7

**Experiments And Results:**

great

**Questions For The Authors:**

- I was surprised by the fact that structured skin tone descriptions outperform descriptive text. Could this be because lighting conditions, shadows, or camera exposure may alter how skin tone is perceived visually, causing image-to-text models to generate incorrect descriptions?
- If the goal is to test whether multimodal input can mitigate bias, wouldn’t it be insightful to test across all vision models (e.g., also include CNN + text)? Perhaps this could have demonstrated how well text features uplift even lower-capacity models.
- Some ambiguity in results presentation- “approximately 4% in AUC and accuracy” lacks context (relative to what? which baseline?) Also, how is the gap defined/ calculated here?
- Have the image-to-text descriptions from Gemini and MONET been clinically validated or reviewed by dermatologists for accuracy in this study? I'm curious to know how accurate or precise these descriptions were

**Strengths:**

- The authors provide a well-organized foundation that frames the study's contributions effectively.
- The study presents a comprehensive experimental design for evaluating various vision models, multimodal integration strategies, and the integration of descriptions of skin lesions. I especially appreciated the comparison of the integration strategies.

**Summary Of The Paper:**

The authors investigate the performance of different vision architectures and multimodal approaches for improving fairness in skin cancer screening, particularly with respect to skin tone (light vs dark) bias. The study first compares three vision-only models (a 16-layer CNN, ViT-B/32, and a hybrid ViT-B/32 with ResNet-26) trained solely on skin lesion images. It then explores whether integrating multimodal models (image + text) using 1) skin tone (light or dark) description, 2) Gemini-generated lesion description, 3) MONET-generated concept scores, and 4) a subset of MONET-generated concept scores can improve the performance disparity across skin groups.

**Weaknesses:**

- It’s not evident to me why authors have decided that exploring a new multimodal system is necessary instead of fine-tuning Gemini or MONET directly for classification. Can authors provide more insights and reasoning into this?
- Clarification of the experiment configurations: In Sections IV.B and V.A, it’s unclear whether MONET’s lesion descriptions include skin tone (skin tone + 48 concepts). If the MONET experiment does include skin tone, why wasn’t skin tone also included in Gemini’s output?
- Given the class imbalance in the dataset (e.g., fewer positive cancer cases), it is recommended to include F1 score as part of the evaluation metrics.

Minor
- Undefined acronyms (terms like SGD are well-known but suggested to be defined when used to improve readability)
- Highlights in Tables 1 & 2 need clear legends or explanation (I think some of them might have been accidentally bolded, for example, "Gap" row for "Accuracy" and "AUC" columns for ViT-B/32 with REsNet-26.
- The phrase “light skin AUC slightly exceeds” in Section V.D. seems to downplay the improvement over the baseline model (since this was even higher than the skin tone embedded model). Please consider rephrasing to reflect the magnitude while still acknowledging dark skin drop-offs.
- Work by Benmalek et al. [3] is cited regarding CNN performance disparities across skin tones. Including the quantification of the disparities they observed (e.g., AUC gap or accuracy delta) would be helpful here.

---

### Official Review · Reviewer_YL18 · 2025-07-17
**Systematic evaluation of multimodal ViTs for skin tone bias mitigation but limited by dataset constraints.**

**Confidence:** 4
**Clarity Of Writing:** good
**Clinical Significance:** great
**Methodological Novelty:** good
**Overall Rating:** 7
**Final Rating:** 7

**Experiments And Results:**

great

**Questions For The Authors:**

1. How would the model perform on external datasets from different institutions or imaging conditions? External validation would strengthen confidence in the generalizability of the fairness improvements.
2. Would the conclusions hold with larger ViT architectures if computational resources were available? This could determine whether the approach scales with model capacity.
3. Can the authors detail computational overhead and inference latency for each integration method? If provided, it would inform deployment feasibility.

**Strengths:**

1. Addresses a clinically relevant fairness issue with measurable performance metrics.
2. Uses a structured experimental design comparing multiple model architectures, text sources, encoders, and fusion techniques.
3. Demonstrates that lightweight textual input (skin tone) can significantly improve fairness in classification without increasing model complexity.

**Summary Of The Paper:**

The paper investigates multimodal Vision Transformers that combine visual features with textual inputs to mitigate skin tone bias in skin cancer diagnosis. The study uses the Fitzpatrick 17k dataset. The authors evaluate different text inputs, including skin tone labels, Gemini-generated descriptions, and MONET-generated lesion concepts, using three text encoders (BERT, Sentence-Transformer, and CLIP) and two integration methods (concatenation and fusion). The primary finding shows that ViT-B/32 with BERT-encoded skin tone labels integrated via fusion achieves balanced performance across skin tones with an AUC of approximately 0.82 for both groups and a minimal performance gap.

**Weaknesses:**

1. The study is limited to a relatively small subset (4311 images) of a single dataset, which may affect generalizability.
2. MONET and Gemini-based inputs underperform but are not deeply analyzed.

---

### Official Review · Reviewer_myTW · 2025-07-17
**Good approach for improving Fairness in Skin Cancer Diagnosis via Multimodal ViTs and Structured Text Integration but need further validation**

**Confidence:** 4
**Clarity Of Writing:** great
**Clinical Significance:** great
**Methodological Novelty:** good
**Overall Rating:** 6

**Experiments And Results:**

great

**Questions For The Authors:**

Could you analyze which tokens or phrases in Gemini outputs led to performance degradation, especially on dark skin tones?

**Strengths:**

1) Tackles a real and urgent issue: racial bias in dermatological AI systems due to the underrepresentation of darker skin tones.
2) Uses ViT-B/32, an efficient backbone, with lightweight text integration and BERT encoding—making the method deployable in constrained clinical environments.
3) Uses per-group AUC, sensitivity, and specificity to measure equity across skin tones, rather than just global performance.

**Summary Of The Paper:**

This paper explores multimodal Vision Transformers (ViTs) to address fairness and accuracy disparities across skin tones in skin cancer detection. The authors leverage structured text (skin tone labels and lesion descriptions) from two image-to-text models—Gemini (general-purpose) and MONET (dermatology-specific)—integrated into a ViT framework. The best-performing model, combining ViT-B/32 with BERT-encoded skin tone labels via feature fusion, demonstrates balanced performance (AUC ≈ 0.82, accuracy ≈ 0.82 for both skin tones), significantly reducing the diagnostic bias between lighter and darker skin tones with minimal computational overhead.

**Weaknesses:**

1) Including 48 lesion concepts from MONET often reduced model performance—likely due to noise and sparsity, but the paper does not analyze whether pruning or weighting concepts would help.
2) While Gemini is tested, its outputs degrade model fairness and performance, but no thorough error analysis is provided to understand failure modes (e.g., hallucinations, irrelevant phrases).
3) Only ~400 dark skin images vs. ~3900 light skin. Although performance gaps are narrowed, generalizability to real-world dark-skin populations is questionable without larger or external datasets.
4) The fusion of image and text representations is done at the classification layer via concatenation or element-wise fusion. This is a late fusion strategy that does not allow the model to learn rich cross-modal interactions during feature learning.
5) The model was not tested under distribution shifts (e.g., different lighting, occlusion, noise), nor were adversarial samples or test-time augmentations explored. This is particularly critical in fairness-sensitive applications—where the underserved subgroups (e.g., dark skin) may be disproportionately affected by such perturbations.